# Enhancing mucosal immunity by transient microbiota depletion

Simone Becattini [1✉], Eric R. Littmann[1,7], Ruth Seok[1], Luigi Amoretti[2], Emily Fontana[2], Roberta Wright[2], Mergim Gjonbalaj[1], Ingrid M. Leiner[1,7], George Plitas[1,3,4,5], Tobias M. Hohl [1,6] & Eric G. Pamer[1,2,6,7]

Tissue resident memory CD8[+] T cells (Trm) are poised for immediate reactivation at sites of pathogen entry and provide optimal protection of mucosal surfaces. The intestinal tract represents a portal of entry for many infectious agents; however, to date specific strategies to enhance Trm responses at this site are lacking. Here, we present TMDI (Transient Microbiota Depletion-boosted Immunization), an approach that leverages antibiotic treatment to temporarily restrain microbiota-mediated colonization resistance, and favor intestinal expansion to high densities of an orally-delivered Listeria monocytogenes strain carrying an antigen of choice. By augmenting the local chemotactic gradient as well as the antigenic load, this procedure generates a highly expanded pool of functional, antigen-specific intestinal Trm, ultimately enhancing protection against infectious re-challenge in mice. We propose that TMDI is a useful model to dissect the requirements for optimal Trm responses in the intestine, and also a potential platform to devise novel mucosal vaccination approaches.

[1] Immunology Program, Sloan Kettering Institute, Memorial Sloan Kettering Cancer Center, New York, NY 10065, USA. [2] Lucille Castori Center for Microbes Inflammation and Cancer, Molecular Microbiology Core Facility, Memorial Sloan Kettering Cancer Center, New York, NY 10065, USA. [3] Howard Hughes Medical Institute, Memorial Sloan Kettering Cancer Center, New York, NY 10065, USA. [4] Ludwig Center at Memorial Sloan Kettering Cancer Center, Memorial Sloan Kettering Cancer Center, New York, NY 10065, USA. [5] Breast Service, Department of Surgery, Memorial Sloan Kettering Cancer Center, New York, NY 10065, USA. [6] Infectious Diseases Service, Department of Medicine, Memorial Sloan Kettering Cancer Center, New York, NY 10065, USA. [7] Present address: Duchossois Family Institute, University of Chicago, Chicago, IL 60606, USA. ✉email: simone.becattini@gmail.com

CD8[+] tissue-resident memory T cells (Trm) provide enhanced protective immunity against pathogens[1,2] and tumors[3,4] by rapidly activating cytokine production and exerting cytolytic functions in infected tissues. Vaccination strategies that facilitate the seeding and long-term residence of CD8[+] T cells may augment host protection in specific tissues.

Although the intestinal tract represents a major portal of entry for infectious agents, specific approaches to enhance CD8[+] Trm accumulation in the gut are lacking, in part because of its poor accessibility to interventions.

Induction of antigen-specific CD8[+] T cell responses has been successfully pursued in clinical trials by parenteral injection of attenuated strains of *L. monocytogenes* (Lm)[5]. This Gram-positive, facultative intracellular bacterium can be readily attenuated and engineered to induce potent CD8[+] T cell responses against antigens of choice. In an attempt to specifically promote mucosal immunity, oral administration of *Lm*-based vectors was evaluated and shown to be safe yet poorly immunogenic[6].

The gut microbiota is intimately linked to the development and function of the host immune system, and it can modulate size and overall composition of the mucosal T cell compartment. In fact, specific bacterial strains isolated from the intestine can promote accumulation of CD8[+] T cells in the intestinal mucosa[7], and CD8[+] T cells induced by rationally designed consortia of commensals enhance protection against orally-acquired infections and tumors[8].

In different settings, however, commensal microbes were shown to blunt host immune responses. For instance, commensals-derived ATP in the intestinal lumen can promote T follicular helper (Tfh) cell apoptosis, and oral administration of a Shigella vaccine strain producing apyrase, an enzyme that hydrolyzes ATP, augments antibody production and overall protection in a mouse model[9]. Thus, modulation of the gut microbiota, including restraint of some of its selected functions, can facilitate achievement of optimal T cell responses.

A key function of the gut microbiota is to provide the host with colonization resistance, i.e. protection from infections via direct or indirect reduction of pathogen load or virulence[10]. In particular, we previously showed that commensal microbes rapidly eliminate Lm from the intestinal lumen of orally inoculated mice and that antibiotic treatment preceding Lm administration, by disrupting colonization resistance, increases Lm expansion in the gut[11].

Based on the above considerations, we hypothesized that temporary inhibition of microbiota-mediated colonization resistance may increase the immunogenic potential of Lm in the intestine and facilitate the generation of larger pools of memory T cells. Here, we describe an immunization strategy (TMDI, Transient Microbiota Depletion-boosted Immunization) that encompasses oral administration of engineered Lm vectors following antibiotic treatment. By producing enhanced chemotactic cues and increased antigen load, TMDI augments the generation of antigen-specific CD8[+] Trm up to 100-fold. Thus, TMDI is a straightforward approach that enhances Trm generation in the gut, providing a useful model to dissect the requirements for induction of robust T cell immunity as well as to develop vaccination strategies targeting the gastrointestinal tract.

## Results

### Microbiota depletion enhances intestinal expansion of *Listeria*.
We previously showed that, by inhibiting colonization resistance, a single oral dose of streptomycin preceding *per os* administration of *Listeria monocytogenes (Lm)* resulted in dramatic intestinal expansion of this bacterium[11]. We decided to investigate in greater detail the kinetics of this phenomenon, and to inquire whether genetic manipulation of Lm, such as expression of exogenous antigens or deletion of virulence factors, would affect its course.

Consistent with our previous results, administration of streptomycin followed by oral inoculation with a low dose ($10^7$ CFUs) of ovalbumin-expressing Lm (LmOVA) led to robust and extended bacterial expansion in the gut lumen (Fig. 1a). Robust intestinal expansion also occurred with an attenuated Lm strain ($\Delta actA$ LmOVA) that was previously reported to be defective for gut colonization[12] (Fig. 1a). In contrast, inoculation of antibiotic-naive mice with higher doses of LmOVA ($1 \times 10^8$ CFUs) resulted in barely detectable amounts of the bacterium in the gut lumen (Fig. 1a). In the presence of an intact microbiota, increasing the LmOVA inoculum from $1 \times 10^8$ to $5 \times 10^8$ CFUs, the highest dose associated with 100% survival, induced a weight loss similar that obtained in mice treated with streptomycin and inoculated with $10^7$ CUFs, but did not enhance Lm expansion in the intestinal lumen (Fig. 1a, b).

Importantly, streptomycin-treated mice inoculated with $\Delta actA$ LmOVA did not display any weight loss, a measure of morbidity (Fig. 1b).

These results confirm that impairment of microbiota-mediated colonization resistance enables even avirulent Lm strains to colonize the intestine, as recently shown elsewhere[13]. However, in the presence of an intact commensal microbiota, achieving high luminal densities of LmOVA requires lethal inocula.

We previously demonstrated that streptomycin impairs the anti-listerial activity of the microbiota only transiently, as 5 days post-treatment mice recover baseline resistance to oral Lm infection[11]. Consistent with this observation, we find that antibiotic administration reduces microbiota density 100-fold 1 day post-treatment with full recovery within 3–5 days (Fig. 1c). Administration of Lm did not impact recovery of microbiota density or composition, suggesting that this pathogen does not perturb the intestinal microbial ecosystem (Fig. 1c, d and Supplementary Fig. 1). Longitudinal 16 rRNA gene sequence analyses revealed a dramatic shift in microbiota composition at early time points following streptomycin treatment, and a substantial recovery of the baseline community structure within 30–45 days (Fig. 1d–f), when the Unifrac distance from baseline microbiota configuration was comparable across mice that had received LmOVA, streptomycin or both (Fig. 1f). These findings confirm the rapid recovery in microbiota density and composition after streptomycin treatment reported elsewhere[14].

Thus, streptomycin treatment transiently depletes the microbiota, providing a narrow time window for enhanced expansion of Lm in the gut lumen and inducing moderate compositional shifts in the microbiota which are mostly unaffected by administration of Lm.

### Listeria expansion augments mucosal CD8[+] T cell responses.
We next set out to investigate the impact of intestinal Lm expansion on the resulting antigen-specific CD8[+] T cell response. In the circulating blood, PBS and streptomycin-pretreated animals infected with different doses of LmOVA or $\Delta actA$ LmOVA displayed similar kinetics and magnitude of OVA-specific CD8[+] T lymphocyte expansion, although we observed a trend toward increased numbers in streptomycin pre-treated mice (Fig. 2a and Supplementary Fig. 3A). The frequencies of central, effector or peripheral memory precursors, as determined by cell surface markers defined elsewhere[15], were comparable across groups at day 9 p.i. (peak of the response) (Supplementary Fig. 2A). In marked contrast, at day 9 p.i. we detected significant differences in the expansion of antigen-specific cells in the intestinal tissue (Fig. 2b and Supplementary Fig. 2B, C). Intravascular staining confirmed that virtually none of the tetramer-positive cells

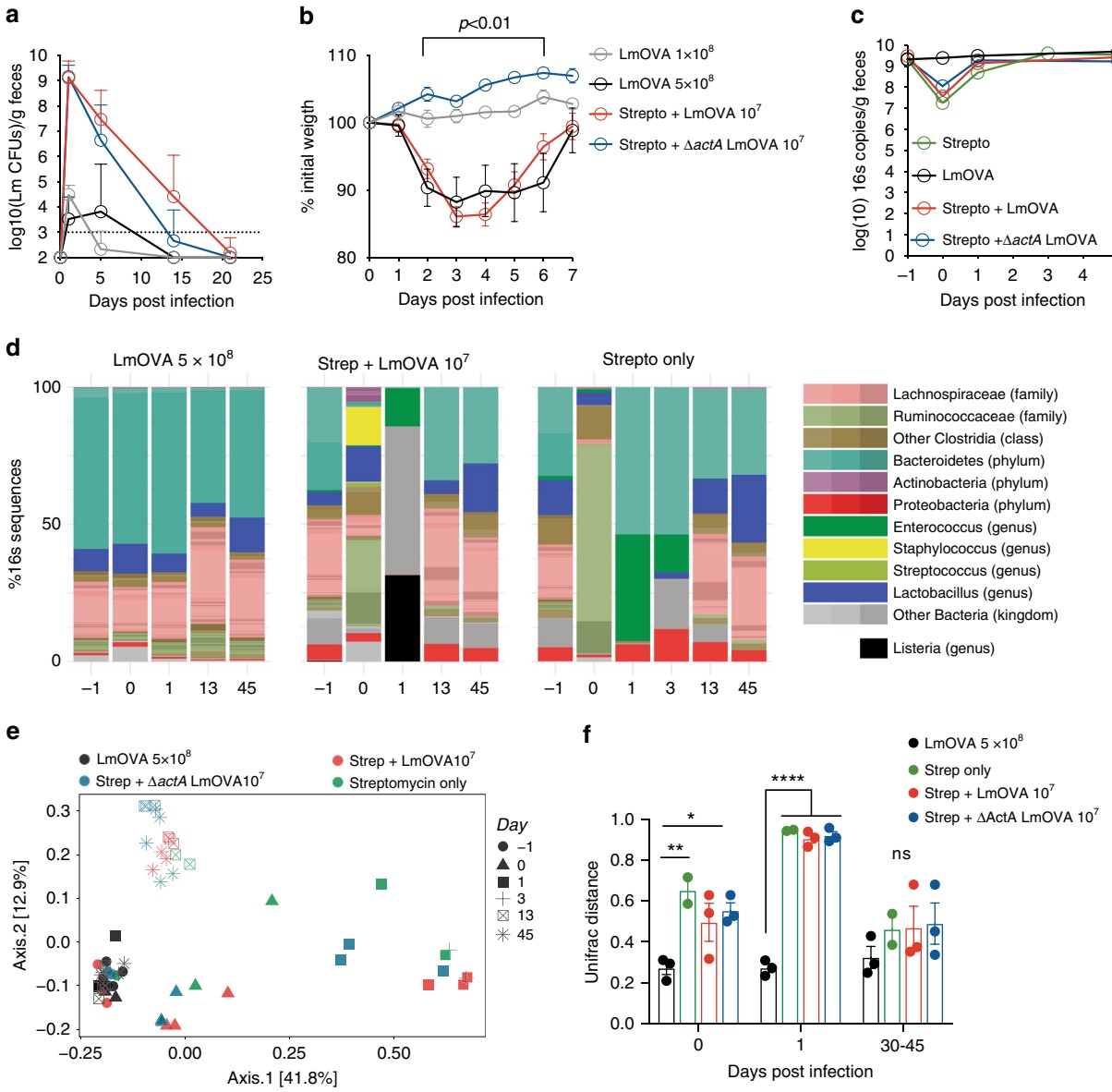

**Fig. 1 Antibiotics promote temporary depletion of the microbiota and transient expansion of Lm in the intestine. a** Fecal shedding of Lm following oral inoculation with different strains performed on mice that had been pre-treated with either PBS or streptomycin at d-1 (for groups listed as in figure: $n = 9$, 10, 15, 10 from 3, 3, 5, and 3 independent experiments, respectively; shown are geometric means ± geometric SD). **b** Weight loss curve for mice treated as shown in **a** (for groups listed as in figure: $n = 9$, 7, 14, 11 from 3, 2, 5, and 4 experiments, respectively; Two-way ANOVA with multiple comparisons, $p < 0.01$, $p < 0.001$ and $p < 0.0001$ at all time points in the indicated time frame between either of the groups: LmOVA $1 \times 10^8$ and Strepto + $\Delta actA$ LmOVA $10^7$ and either of the groups: LmOVA $5 \times 10^8$ and Strepto + $\Delta actA$ LmOVA $10^7$; shown are means ± SEM). **c** Microbiota density in fecal pellets as measured by quantitative 16s gene PCR in mice treated as depicted; 0 refers to the day of infection ($n = 6$ from 2 experiments except d5 $n = 3$ from one experiment, $n = 2$ for Strepto-only group; shown are geometric means). **d** Representative bar graph depicting the composition of the microbiota as assessed by 16s rRNA gene sequencing, for one experiment conducted as described in **a**, **b** (each bar represent pooled data from three mice). **e** Unweighted Unifrac Principal Coordinates Analysis (PCoA) of microbiota composition, based on 16s rRNA gene analysis, for mice treated as depicted in the legend. **f** Overtime Unifrac distance from the baseline microbiota (day −1) obtained from 3 independent experiments performed as the one shown in **d**, **e** (each dot represents one experiment, i.e. average value from 2–3 mice, Two-way ANOVA with multiple comparisons, shown are means ± SEM). (For all analyses: *$p < 0.05$, **$p < 0.01$, ***$p < 0.001$, ****$p < 0.0001$).

detected in the gut lamina propria were located in the circulation at sacrifice while, as expected, a sizable portion of those detected in the spleen were blood-borne (Supplementary Fig. 2D). Streptomycin pre-treatment also significantly increased the proportion of antigen-specific CD69+ cells, particularly of the subset co-expressing CD103, two markers that have been associated with tissue-residency[16] (Fig. 2b and Supplementary Fig. 2C). Thus, antibiotic conditioning enhances early seeding of the intestinal

tissue with antigen-specific CD8+ T that are phenotypically compatible with Trm precursors.

30–50 days post infection (memory phase), a significantly higher proportion of OVA-specific CD8+ T cells was detected in the spleens and intestines of streptomycin- treated animals (Fig. 2c). However, while the increase in spleen was modest and restricted to mice immunized with the virulent LmOVA strain, OVA-specific CD8+ T cells were consistently increased ~10 fold

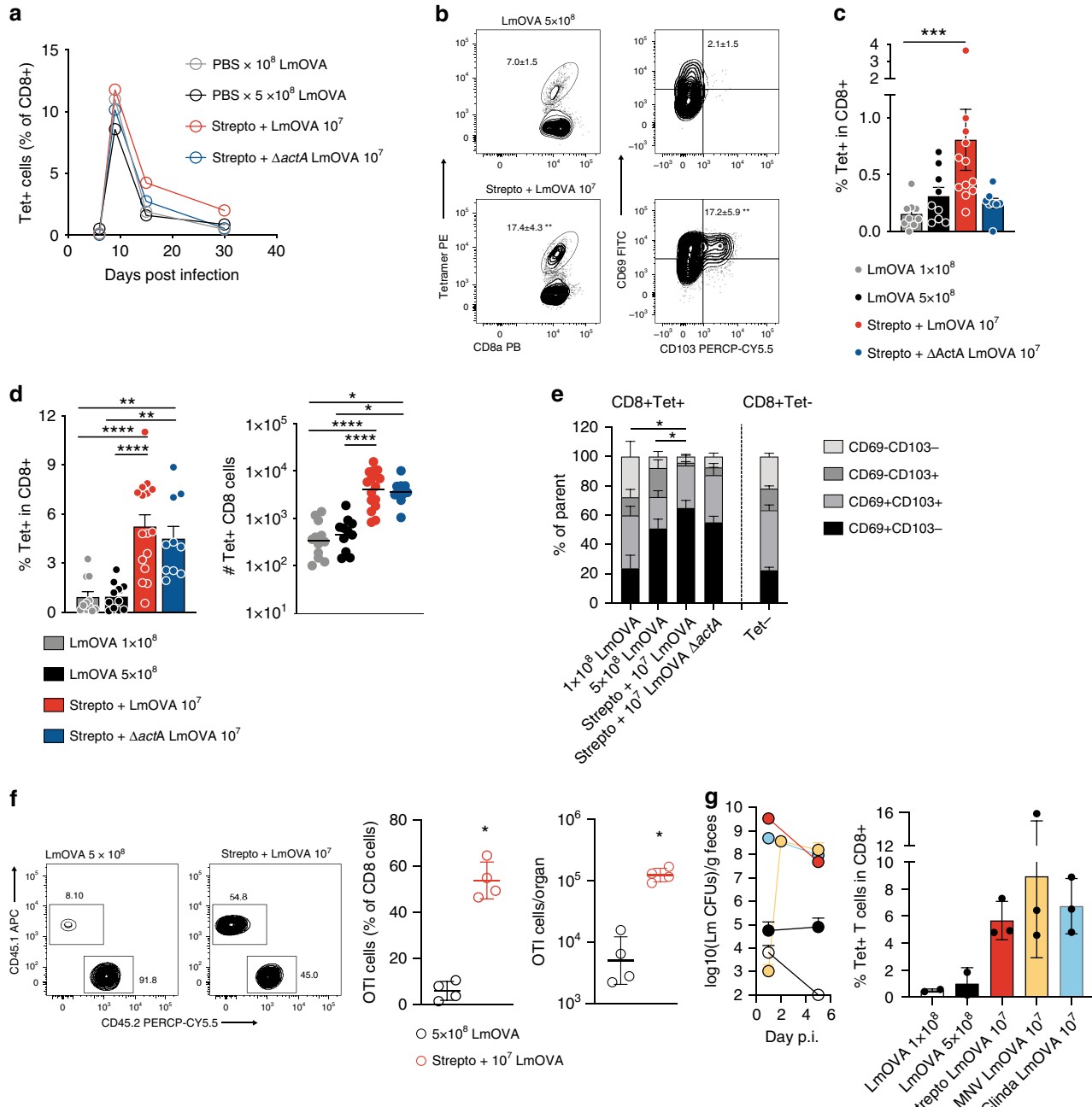

**Fig. 2 Transient microbiota depletion enhances accumulation of antigen-specific CD8$^+$ Trm in the intestine. a** Kinetics of expansion and contraction of SIINFEKL-specific T cells in the blood of immunized animals, as assessed by tetramer staining ($n = 3$, one representative of at least 2 experiments per group; shown are means; see also Supplementary Figure 3 for cumulative data). **b** Representative FACS plots showing percentages and phenotype of tetramer-positive T cells in the large intestine lamina propria at d9 following immunization with the depicted strategies ($n = 6$, data pooled from 2 independent experiments; numbers in plot indicate means ± SEM; Mann–Whitney test). **c** Percentage of tetramer-positive CD8$^+$ T cells in the spleen 30–50 days post immunization (for groups listed as in figure: $n = 9, 9, 12, 8$ from 3, 3, 4, 3 different experiments, respectively; means ± SEM; Kruskal–Wallis test). **d** Percentages and absolute numbers of tetramer-positive CD8$^+$ T cells in the LILP 30–50 days post immunization (for groups listed as in figure: $n = 11, 11, 16, 10$ from 4, 4, 5, 4 different experiments, respectively. One-way ANOVA with multiple comparisons; shown are means ± SEM for percentages and individual values with geometric means for absolute numbers). **e** CD69/CD103 phenotype of tetramer-positive cells in the LILP 30–50 days post immunization ($n = 5$, from 2 independent experiments; one-way ANOVA with multiple comparisons was performed to compare the fraction of total CD69$^+$ cells; shown are means ± SEM). **f** Naïve CD45.1 OTI cells were transferred into naïve CD45.2 C57Bl/6 hosts one day prior to oral administration of LmOVA according to the depicted strategies. Shown are representative FACS plots, percentages and absolute numbers of OTI cells in the LILP 50 days post immunization ($n = 4$, one representative of 2 experiments shown; shown are means ± SD and geometric means ± geometric SD; Mann–Whitney test). **g** Mice were treated once with the depicted antibiotics and 24 h later inoculated with Lm *per os*. Left: Lm CFUs in feces at the depicted time points; shown are geometric means ± geometric SD. Note that low-to-undetectable levels of Lm were shed at d1 by MNV-treated animals, likely the effect of residual antibiotic on Lm growth, but intestinal Lm bloomed on day 2. Right: 50 days post infection mice were sacrificed and antigen-specific cells detected with tetramer. Each dot represents a mouse ($n = 2$–4); shown are means ± SD. A replicate experiment was performed for the clindamycin group, obtaining similar results. (For all analyses: *$p < 0.05$, **$p < 0.01$, ***$p < 0.001$, ****$p < 0.0001$).

in numbers in the intestine of streptomycin pre-treated mice, even when the attenuated ΔactA strain was used for immunization (Fig. 2d).

Following oral LmOVA infection, H2-K$^b$-SIINFEKL-tetramer-positive cells were largely CD69$^+$ (significantly more than tetramer negative CD8$^+$ cells, $p < 0.0001$ for all groups with the exception of $10^8$ LmOVA) (Fig. 2e), and were readily detectable in tissues up to 50 days post infection, indicating that these cells represent bona fide Trm. Within immunized groups, streptomycin treatment produced higher percentages of CD69$^+$ cells (Fig. 2e); CD69$^+$CD103$^-$ cells, which have been associated with protection in a model of intestinal *Y. pseudotuberculosis* infection[17], represented the majority of antigen-specific CD8$^+$ T cells. We used H2-K$^b$-SIINFEKL-specific T cell receptor transgenic OTI cells in an adoptive transfer model and found that streptomycin pre-treatment enhanced expansion of these cells similarly to what observed for the endogenous T cell pool (Fig. 2f).

In models of systemic infection, the magnitude of CD8$^+$ T cell responses to Lm is determined within the first 24 h[18], although it can be boosted by early re-challenge with the same bacterium[19]. To understand the kinetics underlying enhanced memory CD8$^+$ T cell generation in TMDI, we treated mice with ampicillin (which effectively kills Lm within one day[18]) either 24 or 72 h following inoculation. In these settings, allowing for up to 24 h for ampicillin to exert its bactericidal function, viable Lm could only survive for a maximum of 48 or 96 h post inoculation, respectively.

In agreement with previous observations obtained studying splenic responses[18], early Lm depletion did not impair the magnitude or functions of the generated antigen-specific intestinal CD8$^+$ T cells, which were indistinguishable between ampicillin-treated and -untreated animals (Supplementary Fig. 2E–G). Thus, 24–48 h of exaggerated Lm expansion in the lumen might be sufficient to achieve optimal immune stimulation. This indicates that persistence of Lm in the gut lumen after day 2 may not further expand the generated CD8$^+$ T cell pool, possibly because the recovery of the microbiota within this time window decreases the relative density of Lm, impairing its immunostimulatory potential. However, we cannot exclude that viable Lm reservoir survive for longer periods in the intestinal tissue or MLNs despite ampicillin treatment, nor that antigenic-stimulation is still provided by residual, non-viable Lm, as suggested previously[20].

We next investigated whether the observed enrichment in the CD8$^+$ T cell response was specifically dependent on streptomycin treatment, or could be achieved through administration of distinct classes of antibiotics. Strikingly, similar levels of antigen-specific CD8$^+$ T cells were obtained upon administration, prior to immunization, of either clindamycin or a combination of metronidazole, neomycin and vancomycin (MNV) in lieu of streptomycin, suggesting that compounds impairing colonization resistance against Lm are equally suitable to promote immunity CD8$^+$ T cell in these settings (Fig. 2g).

Thus, temporary suspension of microbiota-mediated colonization resistance significantly enhances intestinal CD8$^+$ T cell responses to Lm-encoded antigens. We named this approach Transient Microbiota Depletion-boosted Immunization (TMDI).

**Repeated antibiotic treatments enhance TMDI efficiency.** Given the immunogenic effects produced by LmOVA expansion in the gut lumen, we hypothesized that maintaining or restoring high densities of this vector over time might further increase expansion of CD8$^+$ Trm cells. As microbiota-mediated colonization resistance represents the main mechanism by which Lm density is reduced in the gut lumen[21], we reasoned that repeated administration of streptomycin, to which Lm is resistant, could serve this purpose by impairing recovery of commensal microbes. As predicted, streptomycin gavage at day −1, 4, and 9 with respect to Lm inoculum, resulted in sustained, high-density fecal shedding of LmOVA and induced dramatic accumulation of antigen-specific CD8$^+$ T cells in the LILP, which at day 45–70 post immunization were ~100-fold more abundant than those detected in mice administered standard immunization (Fig. 3a–c and Supplementary Fig. 3A–C). Importantly, mice administered 3 doses of streptomycin did not lose weight upon infection with the attenuated strain *actA*-LmOVA, suggesting that extended TMDI does not induce morbidity (Supplementary Fig. 3D).

Prolonged, dense colonization of the intestinal lumen by LmOVA is likely to result in protracted exposure of T cells to cognate antigen, a circumstance that can promote T cell anergy and exhaustion, including inability to produce effector cytokines[22]. However, TMDI-generated CD8$^+$ T cells appeared to be fully functional, as virtually all detected antigen-specific cells were able to produce IFN-γ upon ex vivo restimulation with cognate peptide (Fig. 3d, e and Supplementary Fig. 3E).

In conclusion, impairment of colonization resistance in the intestinal lumen can be modulated to sustain expansion of antigen-expressing bacteria, and generate higher frequencies of antigen-specific, tissue-resident CD8$^+$ T cells with preserved effector functions.

**TMDI promotes chemotaxis and early T cell recruitment.** The magnitude of the T cell response following TMDI is enhanced at d9 p.i. (Fig. 2b), suggesting that cellular and molecular determinants driving the effectiveness of this strategy are already underway at early time points. In agreement with this hypothesis, using an adoptive transfer model, we detected significantly higher numbers of antigen-specific CD8$^+$ T cells (OTI), but not total CD8$^+$ T cells, in MLNs and LILP of TMDI-immunized animals as early as 4 days p.i. (Fig. 4a and Supplementary Fig. 4A, B). Such enrichment was not detected in the spleen, where we observed a trend toward lower accumulation of OTI cells (Fig. 4a and Supplementary Fig. 4B).

Inflammatory stimuli are crucial for promotion of optimal CD8$^+$ T cell responses in systemic infection[23,24], but their role in intestinal responses is less established. LmOVA TMDI resulted in enhanced transcription of prototypical inflammatory cytokines such as IL-1β and TNF-α in the LILP at day 3 and 6 post inoculation, while a much smaller increase was observed when ΔactA LmOVA was used (Fig. 4b and Supplementary Figure 4C). Consistent with these transcriptional profiles, mice immunized with TMDI displayed proportionally increased influx of myeloid cells, particularly inflammatory monocytes and granulocytes into MLNs and LILP already at d2 post immunization (Fig. 4c–e and Supplementary Fig. 4D). Inflammatory monocytes are a critical component of the innate immune response to *Listeria* infection[25], and produce pro-inflammatory cytokines and nitric oxide. We did not, however, detect reduced CD8$^+$ T cell responses following TMDI in CCR2 KO mice (Fig. 4f), that exhibit markedly reduced inflammatory monocyte due to defective egress from the bone marrow[26–28]. These results are consistent with the observation that inflammatory monocytes, although highly associated with extracellular Lm in the orogastric infection model, do not support intracellular growth and do not serve as antigen presenting cells for this pathogen in the intestine[29]. We conclude that inflammatory monocytes are unlikely to be required for promotion of CD8$^+$ T cells expansion in TMDI.

Importantly, we noticed that chemotactic factors, such as CXCL9 and CXCL10, were also highly up-regulated in the LILP

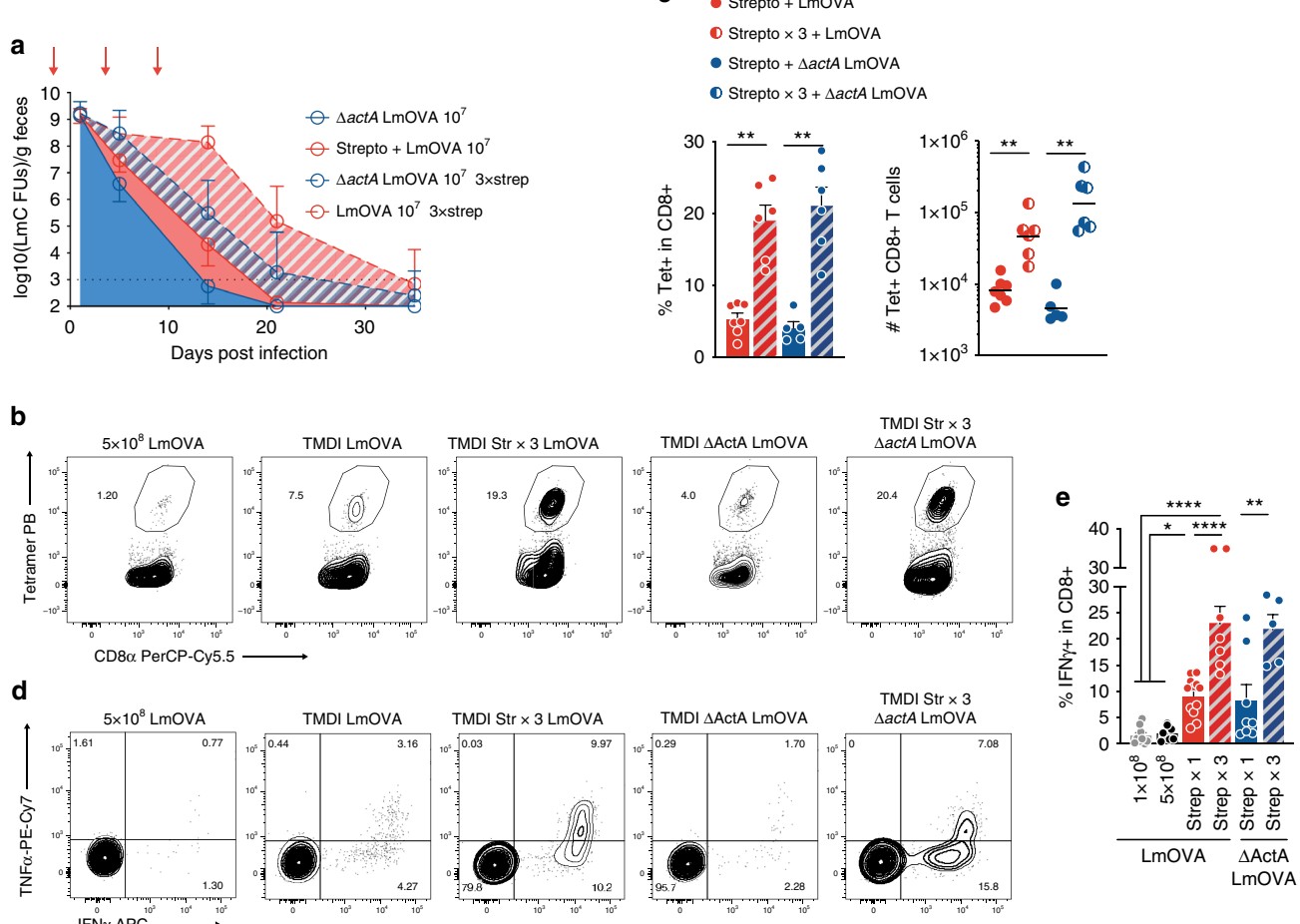

**Fig. 3 Recurrent impairment of microbiota recovery further enhances generation of functional CD8+ Trm. a** Intestinal expansion of LmOVA strains following repeated streptomycin treatments (indicated by arrows, d-1, d4, and d9), as measured by plating of fecal pellets ($n =$ at least 8 from at least 2 experiments per time point per group; shown are geometric means ± geometric SD). **b** Representative FACS plots showing percentage of tetramer-positive cells in the large intestine lamina propria at d50 following immunization with the depicted strategies. **c** Percentages and numbers of tetramer-positive CD8+ T cells in the LILP 50 days post immunization ($n =$ 5–7 from 2 independent experiments; one-way ANOVA with multiple comparisons; shown are means ± SEM and geometric means, respectively. Data points shown for groups administered streptomycin once are a subset of those shown in Fig. 2d). **d** Representative FACS plots and **e** percentages of cytokine production in LILP cells obtained as in **c**, re-stimulated ex vivo in the presence of cognate peptide (SIINFEKL) and Brefeldin-A (for groups listed as in figure: $n =$ 9, 8, 12, 7, 8, 5 from 3, 3, 3, 2, 3, 2 experiments, respectively; one-way ANOVA with multiple comparisons; shown are means ± SEM). (For all analyses: *$p < 0.05$, **$p < 0.01$, ***$p < 0.001$, ****$p < 0.0001$).

and MLNs of TMDI-immunized mice (Fig. 4g, h and Supplementary Fig. 4E, F). These chemokines contribute to CXCR3-mediated recruitment of T cells and Trms to tissues in different settings[17,30], and can be exploited to recruit circulating effector T cells to enhance tissue immunity[2]. Notably, accumulation of antigen-specific CD8+ T cells in the LILP was severely impaired in TMDI-immunized CXCR3 KO mice (Fig. 4i). Our results are consistent with previous findings obtained in a model of oral infection with *Y. pseudotuberculosis*, suggesting a crucial role for the CXCL9/CXCL10-CXCR3 axis in CD8+ T cell migration and localization in the LILP[17,30].

Thus, TMDI enhances production of inflammatory cytokines and chemotactic factors, which can facilitate accumulation of antigen-specific T cells in the LILP.

**Antigen load dictates response magnitude in TMDI.** Besides inflammatory and chemotactic signals, another key factor in promoting efficient priming and expansion of effector/memory CD8+ T cells is antigen load[20,23,24].

In models of systemic infection, antigen availability plays a crucial role in determining the breadth of T cell responses[23,31,32], and strongly contributes to establishment of an immunodominance hierarchy within pathogens' proteomes[20,33].

To ascertain the relative role of antigen load in TMDI, uncoupling it from the influence of inflammatory stimuli, we adopted a strategy to modulate antigen (OVA) availability while maintaining constant the levels of inflammation. Briefly, streptomycin-treated animals were gavaged with a fixed amount of Lm CFUs containing different fractions of antigen-carrying bacteria (100%, 10%, 1% or 0% LmOVA CFUs in total Lm CFUs, the remainder being WT Lm).

At day 1 post infection, overall intestinal burden (total Lm: WT Lm+ LmOVA) was identical across groups; on the contrary, the LmOVA/Lm ratio observed in the fecal pellets changed across groups and mirrored the composition of the respective inoculum, indicating that each Lm strain had expanded proportionally in the intestinal lumen (Fig. 5a).

Consistent with these results, we found no difference in the amount of total CD8+ T cells recovered from the intestine 9 and

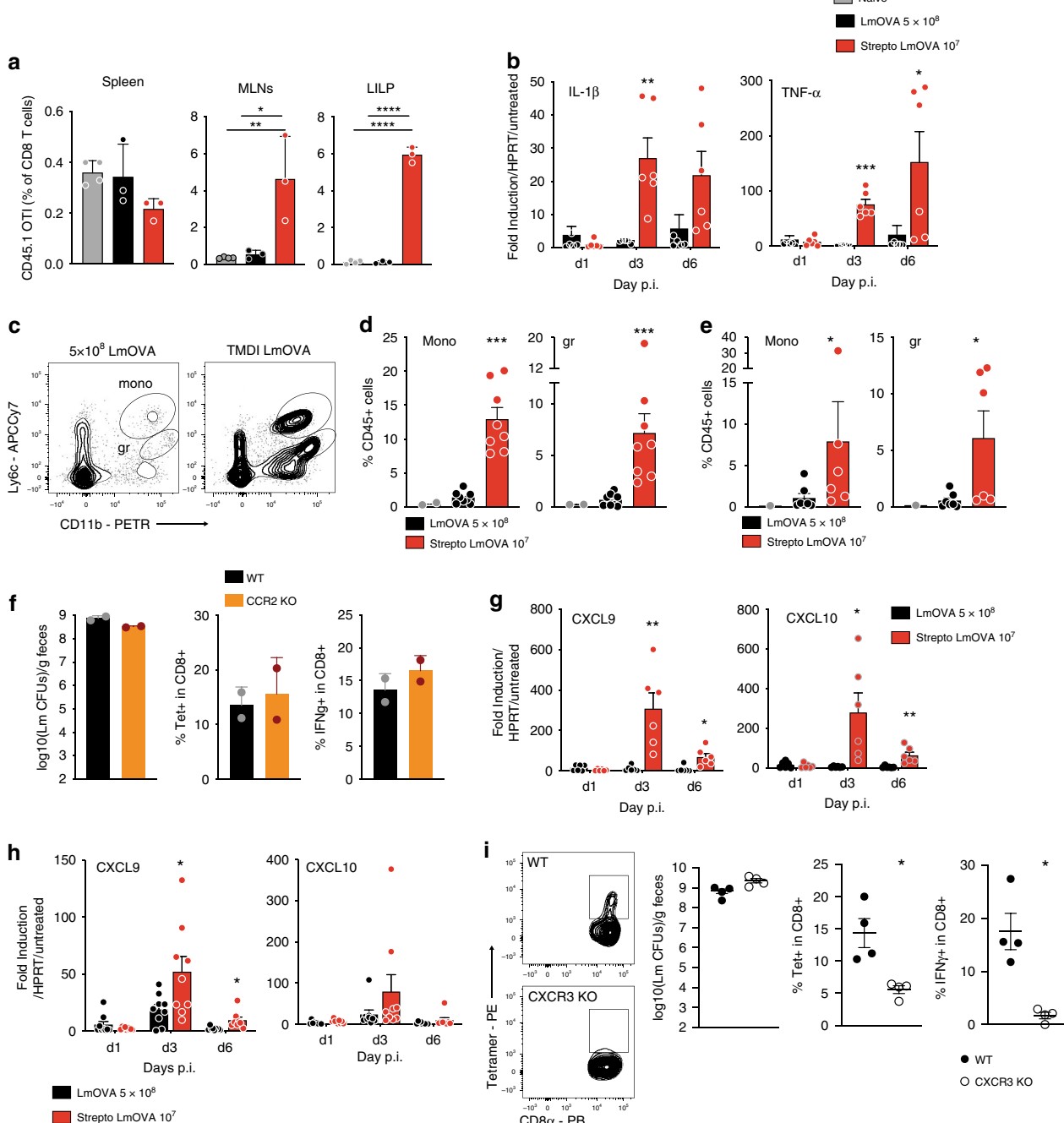

**Fig. 4 TMDI promotes enhanced inflammation and chemotaxis through the CXCL9/CXCL10-CXCR3 axis. a** Percentages of adoptively transferred OTI recovered in different tissues 4 days post immunization with the depicted strategies ($n = 4$, one representative of 2 independent experiments shown; one-way ANOVA with multiple comparisons; shown are means ± SD). **b** qPCR quantification of inflammatory cytokines in the large intestine of mice immunized with standard immunization or TMDI at the depicted time points ($n = 6$ from 2 experiments, shown are means ± SEM, multiple two-tailed $t$-test analysis). **c–e** Representative FACS plots (**c**) and quantification of inflammatory monocytes and granulocytes in the LILP (**d**) and MLNs (**e**) 2 days post immunization with the depicted strategy (**d** $n = 7$ and **e** $n = 8$ from 3 independent experiments; $n = 2$ for untreated mice; shown are means ± SEM; Mann–Whitney test comparing LmOVA 5 × 10^8 and Strep+LmOVA 10^7 groups). **f** TMDI was performed using ΔactA LmOVA in co-housed WT and CCR2 KO mice. Shown are CFUs in fecal pellets at d1 post immunization (left, geometric means ± geometric SD), percentage of tetramer-positive cells in the LILP at d 21, and IFN-ϒ production upon ex vivo restimulation of the same cells in the presence of BFA (right) ($n = 2$, one representative of 3 independent experiments shown, shown are means ± SD). **g, h** CXCL9 and CXCL10 quantification via qPCR in the large intestine (**g**) and MLNs (**h**) of mice immunized as depicted ($n = 6$, data pooled from 2 independent experiments shown, multiple two-tailed $t$-test analysis). **i** Representative FACS plots (left), CFUs in feces at day 1 p.i. (left-center, geometric means ± geometric SD), percentages of tetramer-positive cells in the LILP (right-center) and IFN-γ production upon ex vivo restimulation (right) of the same cells from TMDI (ΔactA LmOVA)-immunized WT and CXCR3 KO mice (d21 post immunization) ($n = 4$ from 2 independent experiments; shown are means ± SEM; Mann–Whitney test). (For all analyses: *$p < 0.05$, **$p < 0.01$, ***$p < 0.001$, ****$p < 0.0001$).

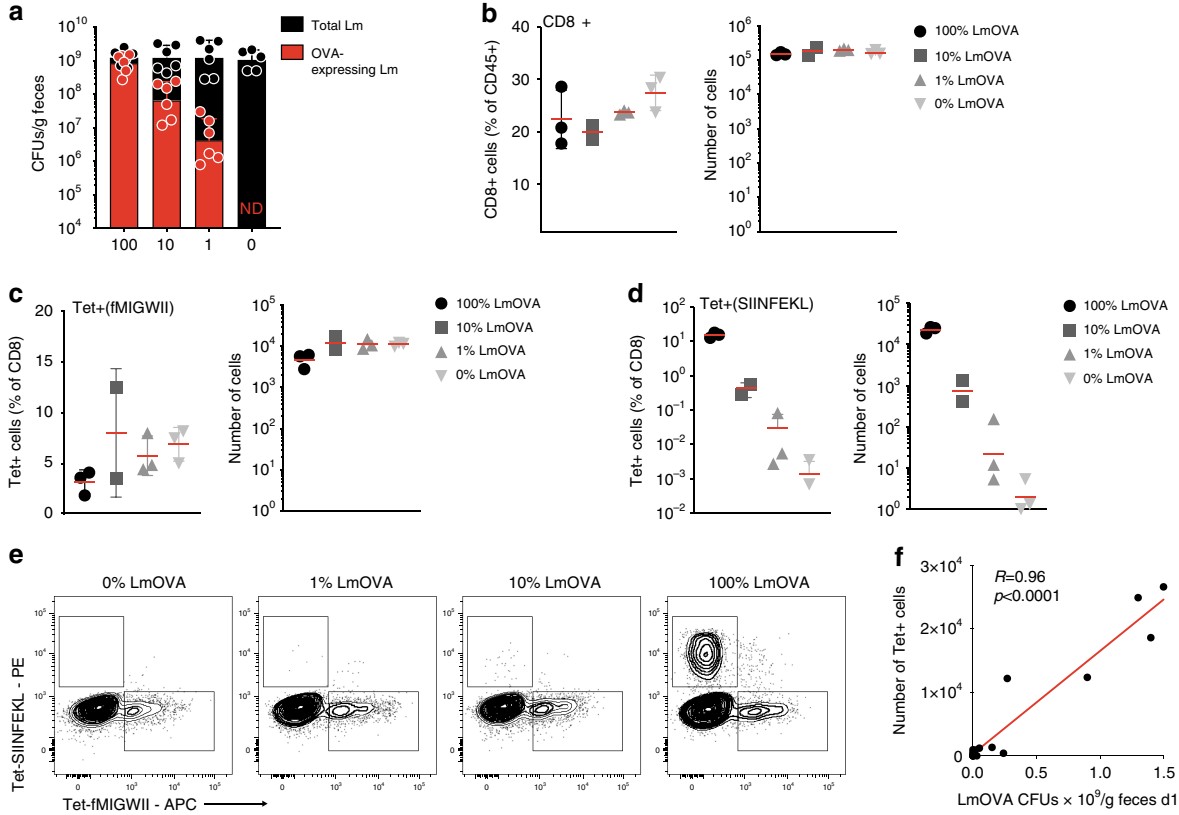

**Fig. 5 High antigen load favors expansion of antigen-specific intestinal CD8+ T cells. a** C57Bl/6 mice were administered streptomycin and one day later infected with a total of $10^7$ Lm CFUs, containing the depicted varying percentages of LmOVA, and a remainder fraction of WT Lm (not expressing OVA). Bars show mean CFUs enumerated for the two Lm strains (WT Lm and LmOVA) in fecal pellets at day 1 post inoculum, obtained through selective plating ($n = 5$-6, from 2 independent experiments; shown are means ± SD). **b, c** Percentage and absolute numbers for total CD8+ T cells and H2-M3:fMIGWII-specific CD8+ T cells in the LILP of mice treated as described in **a**, at day 9 post infection ($n = 2$-3 per group). **d, e** Representative FACS plots and quantification of H2-Kb:SIINFEKL-specific CD8+ T cells in the LILP of animals treated as shown in **a** at day 9 post infection (**b-e**: $n = 2$-3, one representative of 2 experiments shown; shown are means ± SD for percentages and geometric means for absolute numbers). **f** Pearson correlation between number of SIINFEKL-specific T cells in the LILP at day 9 p.i. and LmOVA CFUs/g of feces at day 1 post inoculum in mice treated as described in **a** (r correlation coefficient, and two-tailed are shown in figure). Shown in red is the linear regression curve ($n = 19$ data points from 2 independent experiments).

30 days post infection (Fig. 5b and Supplementary Fig. 5A), nor in the numbers of T cells specific for an antigen encoded by both WT Lm and LmOVA (the fMIGWII peptide, which is presented on the non classical MHC-Ib molecule H2-M3) (Fig. 5c)[34,35]. In contrast, SIINFEKL-specific CD8+ T cell numbers varied greatly across groups (Fig. 5d, e and Supplementary Fig. 5A), strongly correlating with the amount of OVA-expressing bacteria in the intestinal lumen 1 day post inoculum (Fig. 5f and Supplementary Fig. 5B), suggesting that intestinal CD8+ T cell expansion intimately depends upon antigen dose.

Thus, augmenting antigen availability appears to be a key mechanism through which TMDI enhances expansion of CD8+ T cells in the intestine.

**TMDI augments tissue protection.** We next sought to investigate whether enhanced accumulation of memory T cells in the intestinal tissue, particularly CD8+ Trms, achieved through TMDI confers augmented protection to secondary challenge[36]. To this aim, 30 days after receiving either standard oral immunization or TMDI, mice were re-challenged with high doses of LmOVA by oral gavage. Notably, animals immunized with TMDI displayed rapid sterilizing immunity in the intestine as no LmOVA CFUs were retrieved in the tissue at day 3 post infection

(Fig. 6a). In contrast, only modest local protection was achieved in mice previously immunized by oral gavage with $5 \times 10^8$ CFUs of LmOVA (Fig. 6a). TMDI conferred augmented tissue protection even when the attenuated strain ΔactA LmOVA was used (Fig. 6a), a noteworthy finding considering that strains carrying this deletion are currently employed in clinical trials[5,37].

While the present study focused on local expansion of CD8+ T cells, TMDI might impact additional cellular subsets and enhance protection via alternative mechanisms. In fact, defense of the intestinal tissue from orogastric Lm relies upon multiple immune cell types, including CD4+ and γδ T cells, which play partly redundant roles[36,38]. To verify the contribution of different T cell populations to TMDI-mediated protection, immunized mice were re-challenged with high doses of LmOVA per os 30 days post immunization, and concurrently treated with depleting antibodies targeting either CD8+, CD4+ or γδ T cells. Consistent with previous results[36,38], immunized animals were completely protected from re-challenge when either CD8+, CD4+ or γδ T cells were depleted individually, but regained susceptibility when these populations were depleted in combination (Fig. 6b and Supplementary Fig. 6)[38]. Taken together, these results suggest that TMDI might provide augmented immunity by harnessing multiple T cell subsets.

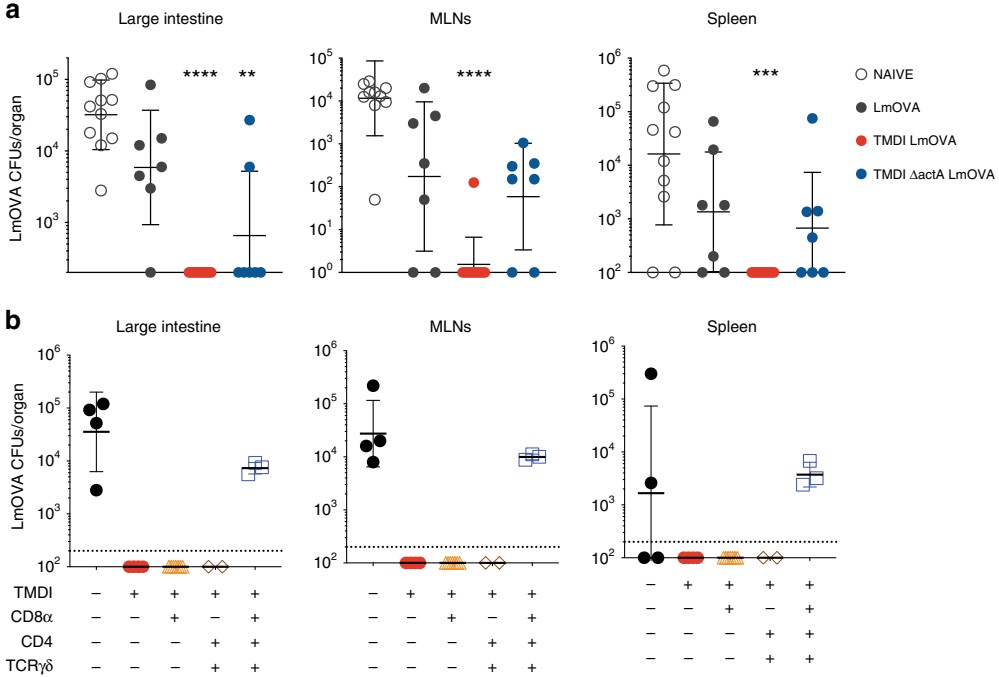

**Fig. 6 TMDI provides enhanced protection and harnesses multiple T cell subsets. a** C57Bl/6 mice were immunized with the depicted strategies and re-challenged orally with $2 \times 10^{10}$ LmOVA CFUs 30 days later. 3 days post re-challenge, mice were sacrificed and LmOVA CFUs in the indicated tissues were enumerated by plating ($n = 5$–9 from 2–3 independent experiments; shown are geometric means ± geometric SD; Kruskal–Wallis test with multiple comparisons, ***$p < 0.001$, ****$p < 0.0001$). **b** C57Bl/6 mice were immunized via TMDI and re-challenged as described in **a**. At day −3, −1, and +1 with respect to re-challenge, mice were injected with the depicted depleting antibodies targeting CD8+, CD4+ or γδ T cells. CFUs were enumerated by plating at d3 post re-challenge ($n = 3$–5 from 1–2 experiments per group; shown are geometric means ± geometric SD).

## Discussion

The intestinal mucosa represents a major portal of entry for pathogens. As a consequence, multiple types of immune cells continuously patrol the intestine, ensuring rapid and effective responses against invading agents[39].

Among such sentinel cells, tissue-resident memory T cells have recently gained center stage in immunological research, due to their extraordinary abundance and crucial protective functions. CD8+ Trm cells, in particular, provide the host with greater protection with respect to their circulating counterparts in multiple models of tumor and infection[16]. Dissecting the mechanistic principles underlying generation and enhancement of intestinal CD8+ Trm is crucial to devise novel tissue-specific vaccines against such diseases[16,40], particularly in the lower GI tract where the developmental pathway and the cellular and molecular requirements for efficient CD8+ Trm induction are much less defined than in tissues such as the skin[17,30,36,41].

In principle, several strategies can be implemented to promote CD8+ Trm priming and accumulation in the intestinal tissue.

Direct delivery of the immunization vector in situ is considered to produce optimal T cell accumulation in tissues and augment protection[42]. Trm generated through local priming are in fact endowed with the ability to rapidly reactivate upon antigen encounter, inducing an alert state in the surrounding tissue, and recruiting subsequent waves of circulating memory T cells[43–45].

Chemokine-mediated recruitment of antigen-specific cells primed in the periphery has also been exploited to selectively enhance seeding of CD8+ Trm in the genital mucosa[2]. In particular, in situ administration of CXCL9 and CXCL10 was used to recruit T cells through the CXCR3 receptor[2], which is rapidly expressed by CD8+ T cells following priming in the setting of infection with intracellular pathogens[15]. Although exogenous administration of chemokines via the oral route to replicate such recruitment in the intestine seems impractical, alternative strategies producing a strong chemotactic gradient toward the gut mucosa might succeed in enhancing local Trm seeding.

Being positioned at the interface with the gut microbiota, the intestinal mucosa encompasses unique immune circuitries[39]. Reactivity toward microbial components of the commensal microbiota must be constantly curtailed, as excessive immune activation is deleterious to the host. Multiple parallel mechanisms are in place to reduce immune responses against the microbiota, including the production of short-chain fatty acids and retinoic acid, which induce peripheral Tregs[46–48], the pro-apoptotic effect of bacterial ATP on intestinal Tfh[49] and the dampening of anti-commensal T cell responses via ILC-mediated antigen-presentation[50].

Additionally, the gut microbiota provides the host with colonization resistance, acting as a first-line defense that can spare the immune system from a direct confrontation with the infectious agent[51].

Thus, despite its vulnerability to infections, the intestine is endowed with multiple safety mechanisms, often microbiota-dependent, that restrain its full activation and make it a rather tolerogenic environment. While these mechanisms are crucial for host homeostasis, they might become unfavorable when aiming at generating optimal immune responses[9].

Here, we present a straightforward approach that maximizes the intestinal T cell response to orally administered engineered Lm strains, by temporarily suppressing the protective activity of the gut microbiota (TMDI, Transient Microbiota Depletion-boosted Immunization). Through administration of antibiotics, the protective commensal species that would otherwise out-compete or actively eliminate Lm are curbed, allowing for

efficient proliferation of the immunization vector. Intestinal expansion of Lm promotes local conditions that ultimately favor the overall accumulation of antigen-specific CD8+ T cells.

In particular, we provide evidence for a major role of both enhanced chemotaxis (CXCL9/10-CXCR3 axis) and increased antigen load in facilitating seeding of the LILP with large numbers of antigen-specific CD8+ Trm.

By repeatedly administering streptomycin after Lm inoculation, thus preventing the recovery of the microbiota and restoring high densities of the vector, we could further expand the resulting pool of CD8+ Trm by one order of magnitude (up to ~30% of OVA-specific cells among total CD8+ T cells in the LILP). This is a remarkable result, particularly as it was obtained by harnessing the restricted endogenous T cell repertoire, rather than exploiting adoptive transfer models.

Importantly, TMDI resulted in better protection from infectious re-challenge, particularly at the intestinal level, as compared to oral immunization of mice bearing an intact microbiota at the time of inoculum. This likely reflects the generation of expanded pools of functional memory cells in the LILP; intriguingly, however, our results suggest that TMDI might harness additional cell subsets beside CD8+ T cells, including CD4+ and γδ T cells, an observation that should be further investigated in future studies.

From an experimental perspective, TMDI represents a unique model to dissect the relative contributions of distinct factors to the generation of optimal intestinal CD8+ T cell responses.

For instance, our study offers novel perspectives on the relationship between antigen-dose and magnitude of the mucosal CD8+ T cell response, a facet that has been rarely addressed in the literature.

By maintaining constant levels of inflammation, we uncovered a direct correlation between antigen dose (as inferred by local expansion of antigen-carrying vector) and size of the generated antigen-specific CD8+ T cell pool. As a recent report suggests that CD8+ Trm accumulation is potently boosted by local antigen presentation in the tissue following priming[52], we speculate that this mechanism might also contribute to antigen-driven TMDI efficiency, given that availability of ovalbumin should be particularly augmented in situ in our model[11].

Overall, these findings might provide useful indications for the rational optimization of mucosal vaccination strategies.

High doses of antigen and sustained inflammatory cues lead to anergy or exhaustion of T cells in a variety of experimental and clinical settings[22]. However, CD8+ Trm generated through TMDI showed no sign of functional exhaustion, as they were capable of producing effector cytokines upon stimulation with cognate peptide, and they provided enhanced protection in vivo. These findings may further support the observation that high antigen dose can promote acquisition of an exhausted phenotype in CD8+ T cells, without necessarily impacting their effector functions[53].

We find that prototypical inflammatory cytokines such as TNF-α and IL-1β are potently induced by LmOVA in TMDI, however, their relative contribution to the expansion of CD8+ T cells in our system remains to be elucidated, as administration of the avirulent strain ΔactA LmOVA in the same conditions yielded an equally robust CD8+ T cell response, in spite of a minor up-regulation of these factors (Fig. 4 and Supplementary Figure 4). Of note, while we mainly focused on quantitative readouts of the immunization, i.e. numbers of generated antigen-specific memory T cells, elegant work has provided mechanistic evidence for qualitative nuances that are promoted by different levels of inflammation, such as distinct levels of antigen affinity in CD8+ T cells[54,55].

The kinetics of TMDI also provides interesting insights onto the biology of intestinal CD8+ T cells. Although TMDI results in prolonged persistence of Lm in the intestinal lumen (weeks), we found that eliminating Lm 24–48 h post inoculum by means of ampicillin treatment, did not impair the generation of an expanded CD8+ Trm pool, in agreement with a previous model of early programming of the memory response[18].

Assuming that ampicillin accomplished rapid and complete clearance of Lm and Lm-derived antigens in our setting, the above findings would seem at odds with the additional immune boost produced through repeated streptomycin treatment.

One possible explanation is that, in order to promote maximal CD8+ Trm priming or boosting, Lm must reach a certain density in the intestine, and that repeated streptomycin treatments magnify the immune response by producing peaks of Lm density that surpass this threshold, thus leveraging a prime-boost mechanism, rather than by sustaining high, but sub-maximal, densities of Lm. In fact, prime-boosting within a 5-day temporal window (also the time span between repeated streptomycin treatments in our model) was found to significantly augment the CD8+ T cell response to Lm administered systemically[19].

As TMDI can promote extraordinary accumulation of functional, antigen-specific Trm in the intestine and augment protection from infectious re-challenge, this strategy should be considered in a translational perspective.

Although allowing deliberate expansion of Lm in the gut lumen poses evident risks, it must be noted that attenuated forms of Lm have been safely administered in clinical trials via parenteral injection, to promote immunity against viral or tumor antigens[5,37,56]. Attenuated Lm strains were also well tolerated by healthy volunteers upon oral administration[6,57].

Multiple strategies have been adopted to minimize the risk of host infection with Lm upon vaccination, including deletion of key virulence factors[58], introduction of suicide plasmids[59] or induction of auxotrophy[60], and a broad range of approaches could therefore be evaluated to ensure safety of TMDI. In addition, our data suggest that ampicillin treatment can be used effectively to rapidly eliminate Lm from the gut lumen in spite of high bacterial burdens.

Encouragingly, using an attenuated Lm strain (ΔactA LmOVA) we showed that maximal accumulation of CD8+ T cell in the LILP could be achieved in the absence of weight loss, a readout for morbidity in the mouse model. Our data confirm recent findings demonstrating that, in the streptomycin-treated gut, avirulent Lm strains can overgrow without inducing any tissue damage or patology[13].

Another important consideration for the advancement of TMDI in a translational perspective is that antibiotic-mediated perturbation of the microbiota can have per se long-term sequelae and predispose to secondary infection[61].

With respect to this concern, strategies to repopulate the intestinal microbiota efficiently are currently being employed in the clinics[62], and consortia of commensal microbes with specific immunomodulatory properties have been established and assembled[8,63].

Our analyses indicate that streptomycin treatment induces a rather limited reconfiguration of the overall microbiota structure, and similar results have been obtained elsewhere using a comparable antibiotic regimen[14]. It could, therefore, be possible, following streptomycin treatment, to restore the original microbial composition, or to engraft a rationally designed consortium of commensal bacteria with features of interest via provision of appropriate probiotics, to curtail the risks of antibiotic-mediated dysbiosis.

Thus, a combination of approaches to minimize the risks of infection with Lm and restore a protective microbiota in a timely manner, might ensure safety of TMDI in a translational perspective.

Enhancing the accumulation of IFN-ϒ producing CD8⁺ Trm in the large intestine in a CXCR3-dependent manner might have important implications for cancer immunotherapy, particularly in the case of colorectal cancer (CRC). In fact, recruitment of CD45RO⁺CD3⁺ T cells in CRC lesions correlates with highly improved prognosis, suggesting that T cells have strong therapeutic potential for treatment of this tumor[64,65]. Furthermore, a recent report showed that the CXCL9/CXCR3 axis, which seems highly stimulated by TMDI, is crucial in determining efficiency of PD1 blockade in a model of colorectal cancer using the MC38 cell line[66].

Importantly, clinical trials have demonstrated that tumor-specific neoantigens can be targeted with therapeutic vaccines[67,68]. Attenuated strains of Lm such as LADD, which is deficient for the virulence factors *actA* and *inlB*[69] are currently utilized in multiple clinical trials[56] and an engineered LADD strain expressing two MC38 neo-epitopes was shown to increase CD8 T cell infiltration into the tumor[58].

In conclusion, TMDI provides a unique model to study the generation of large, functional pools CD8⁺ Trm in the intestine, and represents a conceptually novel approach to harness mucosal immunity against antigens of choice. Further dissection of the molecular events underlying the efficacy of TMDI may provide important insights to the design of improved mucosal vaccines.

## Methods

**Mice.** C57BL/6 mice were purchased from the Jackson Laboratory. $Cxcr3^{-/-}$ and $Ccr2^{-/-}$ mice were purchased from Jackson Laboratories and maintained in house. KO mice were co-housed with WT animals for at least 3 weeks prior to immunization. Unless otherwise specified, 6–12 weeks old female mice were used for all experiments. All animal procedures were approved by the Institutional Animal Care and Use Committee of the Memorial Sloan Kettering Cancer Center.

**Bacterial Strains.** All *L. monocytogenes* strains utilized in this work were derivatives of the 10403s parental strain. *Lm*OVA was generated as described[70]. Frozen aliquots of bacteria were freshly inoculated into BHI and grown to OD 0.1–0.4 (OD = 0.1 corresponds to $2 \times 10^8$ CFUs/ml), centrifuged and resuspended in PBS for inoculation.

**TMDI and antibiotic treatments.** A single dose of streptomycin (ThermoFisher) (20 mg/mouse in PBS) was administered to mice by oral gavage in 200 μl of PBS. 24 h later, mice were inoculated with doses ranging from $10^7$ to $5 \times 10^8$ CFUs of the appropriate *Listeria monocytogenes* strain.

For TMDI experiments shown in Fig. 2g mice were treated with 1 dose of clindamycin (i.p., 200 μg) or a combination of metronidazole, neomycin and vancomycin (per os, 3 mg each) and 24 h later were gavaged with $10^7$ LmOVA CFUs.

For experiments involving ampicillin-mediated Lm killing (Supplementary Fig. 2E–G), 24 or 72 h post TMDI inoculum (i.e. LmOVA gavage following streptomycin treatment) mice received 20 mg of ampicillin by oral gavage and provided drinking water supplemented with 2 g/l of ampicillin. Ampicillin-supplemented water was maintained for the entire duration of the experiment (either 9 or 30 days).

**L. monocytogenes CFU enumeration.** *L. monocytogenes* was administered by oral gavage in all in vivo experiments.

To enumerate *L. monocytogenes* growth in mouse tissue, collected organs were resuspended in PBS Triton X-100 (Fisher) 0.05%, homogenized for 30″ to 1′ with a Power Gen 125 homogenizer (Fisher Scientific) (power level: 5). Metal probes were washed in between samples through 2 immersions in ethanol and one in PBS for 10–15″ each. Serial dilutions of the homogenates were prepared in PBS Triton and plated on BHI plates supplemented with streptomycin (100 μg/ml) and nalidixic acid (50 μg/ml). Colonies were enumerated after 24–36 h of incubation at 37 °C.

For CFU enumeration in intestinal wall, after excision small and large intestine were separated, cleared of content by squeezing with forceps, cut longitudinally and washed vigorously 4–6 times (10 s vortex or manual shaking) in ice-cold PBS. The washed tissues were then processed as described above.

For CFU enumeration in intestinal content and fecal pellets, starting material was weighted (unless total CFU amount was calculated) and resuspended in PBS to a concentration of 100 mg/ml. Serial dilutions of the original suspension were plated.

**Bleeding and tissue preparation.** Blood was obtained by tail bleeding. Red blood cell lysis was performed by 3 consecutive incubations in RBC lysis buffer (0.15 M $NH_4Cl = 1$ mM $NaHCO_3$ in $dH_2O$) for 5′.

Lymphocytes were isolated from the mLNs by mechanical disruption through 100-μm cell strainers.

Single-cell suspensions were obtained from the large intestine lamina propria by longitudinally cutting the tissue and then washing out content in PBS. Intestinal tissues were incubated at 37 °C under gentle agitation in stripping buffer [PBS, 5 mM EDTA, 1 mM dithiothreitol, 4% fetal calf serum, and penicillin/streptomycin (10 μg/ml)] for 30 min. The remaining tissue was digested with collagenase IV (1.5 mg/ml; 500 U/ml) and deoxyribonuclease (20 μg/ml) in complete medium [Dulbecco's modified Eagle's medium supplemented with 10% fetal bovine serum, penicillin/streptomycin (10 μg/ml), gentamicin (50 μg/ml), 10 mM Hepes, 0.5 mM β-mercaptoethanol, and L-glutamine (20 μg/ml)] for 30 min at 37 °C under gentle agitation. Supernatants containing the Lp fraction were passed through a 100-μm cell strainers and resuspended in 40% Percoll. Samples were then centrifuged for 20 min at $600 \times g$.

**Intravascular staining.** Intravascular staining for identification of circulating T cells in the LILP was performed as described elsewhere[71]. Briefly, mice were injected i.v. with 3 μg of anti CD8 antibody (clone 53-6.7) 3 min prior to sacrifice, and then organs were processed as usual.

**Cognate peptide stimulation.** Single cell suspensions obtained from LILP were cultured in complete medium in U bottom 96 well plates in the presence of SIINFEKL $10^{-8}$M (10 ng/ml) and Golgi Plug (brefeldin A) for 4 h at 37 °C. Following stimulation cells were washed and stained for flow cytometry.

**Flow cytometry.** Lymphocytes were counted and subjected to viability staining (Fixable Aqua Dead Cell staining, Life Technologies, #L34957) and subsequently to receptor Fc blockade (BD #553142). Staining was performed using the following antibodies: CD4 (clone RM4-4, BD), CD45 (clone 30-F11 eBioscience), CD45.1 (clone A20, BD), CD45.2 (clone 104, BD) CD8a (clone 53-6.7, eBioscience or BD), CD8b (clone YTS156.7.7, BioLegend), CD11b (clone RM2817, Thermo Fisher), Ly6c (clone AL-21, BD), CD3ε (clone 145-2C11, BD), CD19 (clone 1D3, BD), CD90.2 (53-2.1, BD), CD127 (clone A7R34, BD), CX3CR1 (clone SA011F11, BioLegend), CXCR3 (clone CXCR3-173, Biolegend), CD103 (clone 2E7, eBioscience or Biolegend), CD69 (clone H1.2F3, BD), TCR-β (clone H57-597, BD, TCR-γδ (clone eBio-GL3, eBioscience). H2-K$^b$:SIINFEKL or H2-M3:fMIGWII tetramers were either produced in house or, for the former, obtained thorugh the NIH tetramer core. Samples were fixed (IC Fixation Buffer, eBioscience), washed, resuspended in FACS buffer and acquired with a LSRII flow cytometer (BD) either immediately or on the following day. For intracellular staining, the following antibodies were used: IFN-ϒ (clone XMG1.2, BD), TNF-α (clone MP6-XT22, BD) in Permeabilization buffer (eBioscience).

**Cell transfer experiments.** CD8⁺ T cells cells were obtained from spleen of CD45.1 OTI female mice utilizing the mouse CD8a⁺ T Cell Isolation Kit (Miltenyi Biotec, # 130-104-075) according to manufacturers' instructions. Purity of the cells was confirmed by flow cytometry to be >95%.

$1.5 \times 10^5$ cells were injected i.v. into WT CD45.2 C57Bl/6 mice (Jackson) anesthetized with isofluorane. Whenever coupled with TMDI, OTI administration was performed on day −1, approximately 1 h prior to streptomycin administration.

**Re-challenge and depletion experiments.** Mice were immunized with different strategies using OVA-expressing Lm strains. 30 days following primary immunization, mice were re-challenged by oral gavage with $2 \times 10^{10}$ LmOVA. Mice were euthanized 3 days post infection and harvested organs were homogenized and plated.

For depletion experiments, mice were treated on day −3, −1 and +1 (with 0 being the day of re-challenge) with the following antibodies: α-CD8α (Clone 2.43, 500 μg), α-CD4 (GK1.5, 200 μg) α-γδ TCR (UC7, 100 μg) (all from BioXCell) similarly to what previously published[36]. Mice were sacrificed 3 days post re-challenge and organs plated.

**Tissue RNA isolation, cDNA preparation, and RT-PCR.** RNA was isolated from colon tissue using mechanical homogenization and TRIzol isolation (Invitrogen) according to the manufacturer's instructions. cDNA was generated using Quanti-Tect reverse transcriptase (QIAGEN). RT-PCR was performed on cDNA using TaqMan primers and probes in combination with TaqMan PCR Master Mix (ABI), and reactions were run on a RT-PCR system (StepOne Plus, Applied Biosystems). Gene expression is displayed as fold increase over uninfected control mice and normalized to *Hprt*.

**DNA extraction and 16S rRNA gene sequencing.** To extract DNA from fecal material, a frozen aliquot (~100 mg) of each sample was suspended, while frozen, in a solution containing 500 μl of extraction buffer (200 mM Tris, pH 8.0/200 mM NaCl/ 20 mM EDTA), 200 μl of 20% SDS, 500 μl of phenol:chloroform:isoamyl alcohol

(24:24:1), and 500 μl of 0.1-mm diameter zirconia/silica beads (BioSpec Products). Microbial cells were lysed by mechanical disruption with a bead beater (BioSpec Products) for 2 min, after which two rounds of phenol:chloroform:isoamyl alcohol extraction were performed. DNA was precipitated with ethanol and re-suspended in 50 μl of TE buffer with 100 μg ml$^{-1}$ RNase. The isolated DNA was subjected to additional purification with QIAamp Mini Spin Columns (Qiagen). For each sample, duplicate 50 μl PCR reactions were performed, each containing 50 ng of purified DNA, 0.2 mM dNTPs, 1.5 mM MgCl2, 2.5 U Platinum Taq DNA polymerase, 2.5 μl of 10X PCR buffer, and 0.5 μM of each primer designed to amplify the V4-V5: 563 F (5′-nnnnnnnn-NNNNNNNNNNNN-AYTGGGYDTAAAGNG-3′) and 926 R (5′-nnnnnnnn-NNNNNNNNNNNN-CCGTCAATTYHTTTRAGT-3′). A unique 12-base Golay barcodes (Ns) precede the primers for sample identification[72], and 1-8 additional nucleotides were placed in front of the barcode to offset the sequencing of the primers. Cycling conditions were 94 °C for 3 min, followed by 27 cycles of 94 °C for 50 s, 51 °C for 30 s, and 72 °C for 1 min. 72 °C for 5 min is used for the final elongation step. Replicate PCRs were pooled, and amplicons were purified using the Qiaquick PCR Purification Kit (Qiagen). PCR products were quantified and pooled at equimolar amounts before Illumina barcodes and adaptors were ligated on using the Illumina TruSeq Sample Preparation protocol. The completed library was sequenced on an Illumina Miseq platform following the Illumina recommended procedures with a paired end 250 × 250 bp kit.

The 16S (V4-V5) paired-end reads were merged and demultiplexed. The UPARSE pipeline[73] was used to: (1) perform error filtering, using maximum expected error (Emax = 1)[74], (2) group sequences into operational taxonomic units (OTUs) of 97% distance-based similarity, (3) identify and remove potential chimeric sequences, using both de novo and reference-based methods. Operational taxonomical units (OTU) were classified using a modified version of the Greengenes database[75].

**Statistical and data analysis**. Data are presented as means ± SD/SEM or geometric means ± geometric SD. Analyses were performed using GraphPad Prism version 7.0a or R-3.3.2.pkg. Statistical tests used included: Mann–Whitney test for two group comparisons, Kruskall–Wallis test with Dunn's multiple comparisons for three or more group comparisons, two-way ANOVA for time courses. Significance values are indicated as follow: *$p < 0.05$, **$p < 0.01$, ***$p < 0.001$, ***$p < 0.0001$. PCoA was performed using the "ordinate" function in the R package "phyloseq", with method = "PCoA", distance = "unifrac".

**Reporting summary**. Further information on experimental design is available in the Nature Research Reporting Summary linked to this paper.

## Data availability
Source data for all figures are provided with the paper. 16s rRNA gene sequencing data are available under the NCBI bioproject PRJNA634963. Source data are provided with this paper.

## Code availability
Custom R codes for Unifrac distance calculation, microbiota composition and PCoA are available at [https://github.com/elittmann/becattini-enhancing-mucosal-immunity-by-transient-microbiota-depletion]. Source data are provided with this paper.

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

## Acknowledgements

We would like to thank all members of the Pamer lab for discussion, and Joseph Sun and Andrea Schietinger (MSKCC) for providing important suggestions. S.B. was supported by an Early Postdoc Mobility Fellowship from the Swiss National Science Foundation (P2EZP3_159083) and an Irvington Fellowship from the Cancer Research Institute (no. 49679). This work was supported by the NIH grants AI042135 and P30 CA008748 to EGP.

## Author contributions

S.B. and E.G.P. conceptualized the study, designed experiments, and interpreted all the data. S.B. performed all experiments, with assistance from R.S., M.G., I.M.L., G.P. L.A., E.F., and R.W. performed DNA extraction and library preparation for 16s rRNA gene analyses. E.R.L. analyzed 16s rRNA gene sequencing data. S.B. wrote the paper with input from E.G.P. and T.M.H. E.G.P. supervised and acquired funding for the study.

## Competing interests

E.G.P. has received speaker honoraria from Bristol Myers Squibb, Celgene, Seres Therapeutics, MedImmune, Novartis and Ferring Pharmaceuticals and is an inventor on patent application # WPO2015179437A1, entitled "Methods and compositions for reducing Clostridium difficile infection" and #WO2017091753A1, entitled "Methods and compositions for reducing vancomycin-resistant enterococci infection or colonization" and holds patents that receive royalties from Seres Therapeutics, Inc. All other authors declare no competing interests.
