## [Peer Review File · Nature Communications]

Reviewers' comments:

Reviewer #1 (Remarks to the Author):

The manuscript from Becattini et al studied a new method to enhance immunity by transient microbiota depletion. Authors established a transient environment with less colony resistant from healthy microbiota by single dose streptomycin treatment. In this way, a boosted intestinal immune response, especially for the tissue resident memory CD8+ T cells, was achieved after *L.monocytogenes* (Lm)OVA was introduced,. Lm is able to expand longer in lumen with repeated antibiotic treatment. Both wide type Lm strain and the attenuated Lm strain, which is utilized in clinical trials, was able to display enhanced immunity following secondary high dose challenge. The manuscript is very well written and clearly structured. I only have some detailed questions that need to be clarified.

1. In Figure 1D, why there is no Lm from feces at day 1 in the LmOVA group by sequencing?
2. In Figure 1D and Supplementary Figure 1A, it would be nice to have sequencing from all of the days (Day -1, 0, 1, 13, 30, 45) in order to make the story consistent. In addition, because Strep+ Δ ActALmOVA is an important group, it's better to show the data of the Strep+ Δ ActALmOVA group in Fig.2B,3B,3D,4 too.
3. Could you please explain why the Strep only group did not restore the microbiota as good as the Strep+LmOVA group?
4. Figure 3A, it would be interesting to see the weight loss in the four groups as showed in Figure 1B. Did all the them gain the weight back? It is listed data not shown in the current version.
5. In Figure 4B and 4G, both experiments are carried out by qPCR, why B only used one result from two experiments, while G used the results from two experiments? Please include both experiments in B to make things consistent.
1. Figure 5A, it is hard to draw conclusion with day 1 feces without following up days. The %LmOVA stains detected in the feces may be the stains inoculated but failed to colonize in the intestinal lumen. The %LmOVA could be checked by CFU in intestine at day 9.
6. Supplementary Figure 4B, the dots in this figure does not look like a linear correlation but are decreasing as the CFUs are increasing.
7. Overall, there are some inconsistency between figures, e.g authors used day 9 and day 30-50 in Figure B; day 2, day 4 and day 21 in Figure 4; day 9 in figure 5, is there any explanation for not having consistent measuring points in the figures?
8. Similar with previous comment, the numbers of mice vary, e.g n=9-12 from 3-4 experiment from Figure 2C, D and n=8-16 from 2-5 independent experiments from Figure 3A. Please state the exact number of mice used and the exact number of independent experiments for each figure panel. If there are mice not included in the figures, please explain why they are excluded.

There are many other small mistakes:

1. Line 139, one 'antigen-specific' should be deleted.
2. Line 183, addition, the word "delay" is not accurate for Δ ActALmOVA 3xstrep group since there is no Lm in feces from day 20. This is the same date as other groups.
3. Line 206, it is fig.2B, instead of fig.2C, that shows T cell response enhanced at d9.
4. Line 209-210, data of total CD8+ T cells were not shown in figure 4A.
5. The figures numbers and text are not matched from line 259 to 269.
6. Line 543, (G) should be changed to (B).

Reviewer 2 (Remarks to the Author):

Becattini and colleagues combine oral streptomycin and oral infection with strep resistant Ova-expressing *Listeria monocytogenes* (Lm) to show that treatment transiently decreased normal gut flora, resulted in elevated and extended gut Lm loads and enhanced Ova-specific CD8 T cell Trm responses and protective immunity against Lm Ova oral challenge when compared to mice immunized without strep treatment. Of note, as observed by others, enhance protection was not solely dependent on CD8 T cells, but T cells in general. Implications are the potential to improve gut Trm generated by oral vaccination with live bacterial vectors.

The experiments were generally well designed and most of the data robust, I have only one experimental question for the authors. The observation that the CD8 T cell response elicited is dose dependent raises the question of how much the prolonged duration of gut Lm load contributes to the elevated CD8 T cell response. This is particularly an issue with the contention that multiple strep treatments enhance the CD8 T cell response (but see comment 3 below). This notion would be reasonably simple to test, using a second antibiotic to clear the Lm, administered within a few days after initial infection and then measure the T cell response.

Minor comments

1. Please define PCoA somewhere
2. It is unclear why the IV exclusion approach was not employed to verify that the analyzed cells were in tissues. Please justify.
3. The experiment in Fig. S2A evaluates the phenotype of CD8 T cells at day 9 post infection. I would caution the authors not to call these cells memory, no matter what their marker profile.
4. Figure 3C and D, either the groups are out of order or the legend is out of order. In either case, there is no statistical support for the notion that the 3X strep treatment provided a better CD8 T cell response. This conclusion should be tempered or statistical support provided.
4. Quite a few typos remain and there are some sections where the lack of precise description of the experiments/results made the paper a difficult read.
5. The authors may wish to spend a little more emphasis on the need to find attenuated bacterial delivery systems for translation.
6. Would this work with other antibiotics and Lm resistant to same or is it Strep-specific?

Rebuttal letter

Becattini et al., ' Enhancing Mucosal Immunity by Transient Microbiota Depletion'

We thank the Reviewers for their careful reading of our manuscript and for their constructive comments. We are now re-submitting a highly improved version of our initial work, in which we have addressed all of the raised concerns and added crucial controls as well as novel experiments. We have also thoroughly checked for typos and rephrased unclear sentences to improve clarity for the reader. Please find below a point-by-point response to the Reviewers' comments.

Reviewer #1 (Remarks to the Author):

The manuscript from Becattini et al studied a new method to enhance immunity by transient microbiota depletion. Authors established a transient environment with less colony resistant from healthy microbiota by single dose streptomycin treatment. In this way, a boosted intestinal immune response, especially for the tissue resident memory CD8+ T cells, was achieved after *L. monocytogenes* (Lm)OVA was introduced. Lm is able to expand longer in lumen with repeated antibiotic treatment. Both wide type Lm strain and the attenuated Lm strain, which is utilized in clinical trials, was able to display enhanced immunity following secondary high dose challenge. The manuscript is very well written and clearly structured. I only have some detailed questions that need to be clarified.

1. In Figure 1D, why there is no Lm from feces at day 1 in the LmOVA group by sequencing?

1. The Reviewer points out correctly that *Listeria* is not visible in the bargraphs obtained from 16s rRNA gene sequencing. This is in agreement with the low expansion of *Listeria* at d1 post infection in the absence of streptomycin conditioning, which is about 10^4 CFUs/g feces (Figure 1A). Considering an approximate bacterial load in the intestine of at least 10^9 - 10^{10} bacteria/g feces (widely reported in the literature, see also qPCR data in Figure 1C) the above levels of *Listeria* correspond to less than 0.001% of the total, an amount too small to be even detected with the utilized sequencing depth).

2. In Figure 1D and Supplementary Figure 1A, it would be nice to have sequencing from all of the days (Day -1, 0, 1, 13, 30, 45) in order to make the story consistent.

While we agree that having identical time points would have been optimal, unfortunately fecal pellets were collected at different time points for the depicted experiments. To increase consistency we sequenced pellets from one more experiment that had been performed with time points d-1, d0, d14, d40 which very closely approximate what shown for one of the two previously included experiments. We have now included a summary in Figure 1 that incorporates data from the 3 replicate experiments, and cumulated all data from the later time points into a 30-45 days group, so to make the data consistent with the subsequent T cell analyses.

In addition, because Strep+ Δ ActALmOVA is an important group, it's better to show the data of the Strep+ Δ ActALmOVA group in Fig.2B,3B,3D,4 too.

We agree with the Reviewer that these data are important and we have complemented the indicated figures with panels depicting the Strep+ Δ ActALmOVA group in Supplementary Figure 2, 3 and 4.

3. Could you please explain why the Strep only group did not restore the microbiota as good as the Strep+LmOVA group?

We believe the data shown in Figure 1D and 1E argue instead that there was no significant difference between the Streptomycin only group and the Streptomycin + Lm groups, as also quantified in Figure 1E, right panel. If anything, streptomycin treatment alone perturbed the microbiota to a lesser extent, which is to be expected given that the Strep + LmOVA group is administered 2 perturbing stimuli instead of 1.

4. Figure 3A, it would be interesting to see the weight loss in the four groups as showed in Figure 1B. Did all of them gain the weight back? It is listed data not shown in the current version.

We agree with the Reviewer that this piece of data is informative and we have added it to the manuscript (Supplementary Figure 3D).

5. In Figure 4B and 4G, both experiments are carried out by qPCR, why B only used one result from two experiments, while G used the results from two experiments? Please include both experiments in B to make things consistent.

The Reviewer points out correctly that only one of the two repetitions was included here, in the corrected figure cumulative data from 2 independent experiments are shown.

6. Figure 5A, it is hard to draw conclusion with day 1 feces without following up days. The %LmOVA strains detected in the feces may be the stains inoculated but failed to colonize in the intestinal lumen. The %LmOVA could be checked by CFU in intestine at day 9.

The Reviewer points out correctly that persistence of higher CFUs at later time points might have an impact on the accumulation of antigen specific T cells. However, following the suggestion of Reviewer #2, we have added data showing that Ampicillin treatment of mice 24h post infection, which effectively eliminates intestinal Listeria by day 2, does not impact the size of the antigen-specific T cell population generated with TMDI. Thus, it appears that initial Lm expansion in the gut lumen is sufficient to enhance CD8+ T cell generation, and therefore it is safe to assume that the differences in luminal Lm expansion detected at d1 post inoculum are a good predictor of Trm accumulation at later time points.

7. Supplementary Figure 4B, the dots in this figure does not look like a linear correlation but are decreasing as the CFUs are increasing.

We agree with the Reviewer that plotting the data on a linear scale might be misleading, and therefore we are providing a modified graph with values plotted on a log scale (undetected CFUs in the Lm-only group have been arbitrarily expressed as =1 to allow placement onto the log scale). Please note that although there is a small decrease in the amount of T cells with the increase of the counted CFUs at highest LmOVA concentration, those variations are minimal (2-3 fold) while a decrease in Lm CFUs by one or multiple orders of magnitude resulted in decrease in T cells by orders of magnitude too. Therefore we stand by our conclusions, which have now been corroborated by a linear regression analysis indicating that the data fit onto a curve with equation: $Y = 1.517e-005 * X + 2784$ ($p < 0.0033$).

8. Overall, there are some inconsistency between figures, e.g authors used day 9 and day 30-50 in Figure B; day 2, day 4 and day 21 in Figure 4; day 9 in figure 5, is there any explanation for not having consistent measuring points in the figures?

The Reviewer is correct in that different time points were chosen. We believe that the choice of these different time points served the purpose of dissecting a dynamic scenario that involves multiple cell types at different stages.

In particular, d9 and d30-50 were used as representative of effector phase (peak response) and memory phase, respectively, for T cells.

For qPCR analyses, d1, 3, and 6 were chosen in an attempt to span the duration of the early phase precluding at the T cell peak.

d2 was chosen for analyses of myeloid cells as this was shown elsewhere (see Johnes & D'Orazio, JI, 2017) to be the time required for myeloid cells to migrate to the intestinal LP and MLNs and also in light of the high levels of cytokines likely of myeloid origin detected in those tissues at d3 p.i.

As for OTI accumulation, in time course experiments d4 resulted the earliest time point at which transgenic T cells could be detected in mlns and LILP.

9. Similar with previous comment, the numbers of mice vary, e.g n=9-12 from 3-4 experiment from Figure 2C, D and n=8-16 from 2-5 independent experiments from Figure 3A. Please state the exact number of mice used and the exact number of independent experiments for each figure panel. If there are mice not included in the figures, please explain why they are excluded.

We apologize for the confusion our captions may have caused.

No mice were excluded from the analyses in these plots, the differences in final numbers reflect the following:

1) different experimental groups contained different numbers of mice (usually 2-5 mice per group per experiment);

2) due to the large number of experimental groups utilized, not all experimental groups were included in all experiments (for instance, for OVA-specific T cell enumeration the LmOVA 1×10^8 group was only included in 3 out of 5 experiments performed, while the Strep + LmOVA was present in all 5 experiments);

3) in time point experiments not all time points were analyzed in each experiment (for instance, when assessing kinetics of CD8+ T cells in the blood, analysis of the d6 time point, which precedes expansion of the T cells, has been repeated less times than the that of the d9 time points, for which blood was collected also from animals

sacrificed at d9 for spleen and LILP harvest, in an attempt to increase the observations for the most relevant/variable points).

The same applies to fecal pellet-related plots (such as that shown in Figure 3A) in which additionally some data points are occasionally missing due to the impossibility to obtain a fecal pellet from a specific mouse on a given day.

We have now updated all figure captions to incorporate the number of replicates/experiments. However, for time course experiments (Figure 3A and Supplementary Figure 3A), we are only indicating the range of n/experiments to avoid listing of n for each time point in each group, which would result in rather confusing captions. We believe that given the amount of experimental points presented and the magnitude of the differences detected, indicating the overall range of experimental points/experiments should not detract from the validity of our conclusions.

There are many other small mistakes:

1. Line 139, one 'antigen-specific' should be deleted.

The mistake was corrected

2. Line 183, addition, the word "delay" is not accurate for Δ ActALmOVA 3xstrep group since there is no Lm in feces from day 20. This is the same date as other groups.

We are providing an updated Figure 3A containing 1 more experiment showing that Lm was shed also from Δ ActALmOVA 3 x strep-treated mice at d21. Regardless, even when looking at the previous version of this figure, it should be noted that higher levels of luminal Δ ActALmOVA were achieved at early time points when streptomycin was administered 3 times (compare filled blue line, Δ ActALmOVA, with striped blue line, Δ ActALmOVA 3 x strep, at d6 and 14 in Figure 3a). As the microbiota represents the main mechanism reducing luminal Listeria burden at these time points, we believe it is fair to conclude that a 'lack of' or 'delay in' microbiota recovery is occurring in these mice, with consequent failure to exclude Lm from the lumen. The referenced paragraph has anyway been rearranged in the final version of the manuscript and there is no mention of delay in its current form.

3. Line 206, it is fig.2B, instead of fig.2C, that shows T cell response enhanced at d9.

The mistake was corrected

4. Line 209-210, data of total CD8+ T cells were not shown in figure 4A.

The Reviewer points out correctly that we mistakenly omitted the data we are referring to in this paragraph; we have now added the data in Supplementary Figure 4A.

5. The figures numbers and text are not matched from line 259 to 269.

The mistake was corrected.

6. Line 543, (G) should be changed to (B).

The mistake was corrected.

Reviewer 2 (Remarks to the Author):

Becattini and colleagues combine oral streptomycin and oral infection with strep resistant Ova-expressing *Listeria monocytogenes* (Lm) to show that treatment transiently decreased normal gut flora, resulted in elevated and extended gut Lm loads and enhanced Ova-specific CD8 T cell Trm responses and protective immunity against Lm Ova oral challenge when compared to mice immunized without strep treatment. Of note, as observed by others, enhance protection was not solely dependent on CD8 T cells, but T cells in general. Implications are the potential to improve gut Trm generated by oral vaccination with live bacterial vectors.

The experiments were generally well designed and most of the data robust, I have only one experimental question for the authors. The observation that the CD8 T cell response elicited is dose dependent raises the question of how much the prolonged duration of gut Lm load contributes to the elevated CD8 T cell response. This is particularly an issue with the contention that multiple strep treatments enhance the CD8 T cell response (but see comment 3 below). This notion would be reasonably simple to test, using a second antibiotic to clear the Lm, administered within a few days after initial infection and then measure the T cell response.

We thank the Reviewer for this very important suggestion.

Initial attempts to perform this experiment by treating TMDI-immunized mice with ampicillin (that kills *Listeria*) failed, as Lm burden was rapidly regained shortly after. This is consistent with our previous report (Becattini et al., JEM 2017) showing that as little as 100 Lm particles can fully colonize the intestine of an antibiotic-treated mouse. It is easy to imagine that a few residual Lm CFUs in the gallbladder, intestine or even bedding might have driven the rebound in these conditions.

We therefore resolved to administer mice ampicillin ad libitum in drinking water (together with a one-time oral gavage) starting at either 24h or 72h post inoculation and for the entire duration of the experiment, to ensure that the intestine would remain Lm-free.

Our data, now presented in Supplementary figure 2 F-H, suggest that the initial expansion achieved by LmOVA during the first 24-48h should be sufficient to obtain a full-size TMDI response, as assessed through quantification of OVA-specific CD8+ T cells both at d9 and d50 post treatment (effector/memory phase).

We believe the observation that repeated streptomycin treatments further (and potentially) boost such expansion is not conflicting with the above findings, but rather hints at a certain density of *Listeria* as being critical to obtain boosting effect.

In fact, following TMDI, the density of *Listeria* rapidly decreases, as the microbiota recovers largely by day 3, likely causing a drop not only in the absolute counts of luminal *Listeria*, but also in relative Lm representation within the microbiota. It is reasonable to assume that such diminished density may not be impacting much further the expansion of recently primed T cells. However, each repeated streptomycin treatment brings back the density of *Listeria* in the gut lumen to levels of $\geq 10^9$ /g feces, which are likely promoting a boost in the accumulation and proliferation of the antigen-specific CD8 T cells. We therefore speculate that the high Lm densities achieved through TMDI can prime a full CD8 T cell response

within less than 48h, and that repeated TMDI can boost such expansion by bringing Lm densities back to immunogenic levels and promoting prime-boost effects. We believe that these experiments uncovered a fascinating aspect of the proposed approach, which will need to be further clarified in subsequent work.

Minor

comments

1. Please define PCoA somewhere.

The Reviewer is correct that the acronym had not been described in the text. We have corrected this mistake and Figure 1 caption now indicates Principal Coordinate Analysis as the full name for PCoA.

2. It is unclear why the IV exclusion approach was not employed to verify that the analyzed cells were in tissues. Please justify.

While i.v. exclusion is considered very good practice in the assessment of tissue-resident memory T cells at multiple anatomical location, it is rarely utilized in studies focusing on the intestinal tissue, under the assumption that circulating cells minimally affect the T cell pool retrieved at this location.

We also reasoned that the differences in magnitude of CD8 T cell pools observed across groups must exclusively rely on tissue-resident cells as no major differences were observed in the blood of streptomycin treated or untreated animals. Furthermore, at later time points (d30-50), the % of antigen-specific CD8 T cells in the blood was rather low as compared to that retrieved in the LILP of TMDI-treated mice, making it unlikely that circulating cells could contribute to such pool.

Since i.v. exclusion limits the capacity to process large number of organs and can be a rather expensive procedure, in the light of the above considerations we had initially disregarded the possibility to perform it.

However, as we recognize that neglecting this control might have weakened our conclusions, we have performed one i.v. experiment in which ActA-LmOVA or ActA-LmOVA TMDI-immunized mice were stained for circulating CD8 T cells 3 minutes prior to sacrifice at d9 post infection, when the amount of circulating T cells is the highest (Supplementary Figure 2D).

The analysis confirmed that, while a substantial amount of antigen-specific T cells detected in the spleen resulted to be blood-borne at this early time point, only a minor fraction of the Ag-specific cells detected in the intestine were of circulating origin (<3%), supporting the idea that the T cells detected in the LILP throughout our experiments are bona fide Trm.

3. The experiment in Fig. S2A evaluates the phenotype of CD8 T cells at day 9 post infection. I would caution the authors not to call these cells memory, no matter what their marker profile.

We completely agree with the Reviewer's comment. This mistake was corrected, and now the main body states "The frequencies of central, effector or peripheral memory T cell precursors...".

4. Figure 3C and D, either the groups are out of order or the legend is out of order. In either case, there is no statistical support for the notion that the 3X strep treatment

provided a better CD8 T cell response. This conclusion should be tempered or statistical support provided.

The figure has been updated to provide clearer representation of the data and it has been complemented with statistical analysis supporting the claim that Strep x 3 treatment further magnifies the effects of TMDI (about 10X in absolute numbers).

4. Quite a few typos remain and there are some sections where the lack of precise description of the experiments/results made the paper a difficult read.

We apologize for the multiple typos and have thoroughly assessed the manuscript, hopefully enhancing its readability.

5. The authors may wish to spend a little more emphasis on the need to find attenuated bacterial delivery systems for translation.

We agree with the Reviewer and we have now emphasized this crucial concept in the discussion section.

6. Would this work with other antibiotics and Lm resistant to same or is it Strep-specific?

We thank the Reviewer for suggesting this relevant experiment.

Based on the results obtained in mice treated with one dose of clindamycin or metronidazole+neomycin+vancomycin prior to administration of LmOVA, now presented in figure 2G, we propose that any antibiotic treatment impairing colonization resistance against Lm is suitable to perform TMDI. Note that Lm is sensitive to both the alternative antibiotic regimens utilized, and that in the case of MNV, Lm was even unable to expand on d1 (likely the result of residual antibiotic-mediated killing) but caught up on day 2 with exaggerated expansion and ultimately induced high levels of CD8 T_{rm}.

REVIEWERS' COMMENTS:

Reviewer #1 (Remarks to the Author):

This is a very nice and interesting manuscript and now it reads very clear and logical to me. I am happy with all the responses to my questions and highly recommend it to be published. There are only minor things that might need to be updated: Line 68, "Shigella" is not normally consider to be a commensal microbe. Line 845, it is better to write "16S rRNA gene sequencing". Other things such as figure 1a to figure 1A in line 106 could be fixed during proof reading.

Becattini et al., Nature Communications

Point-by-point Response to Referees

We sincerely thank Reviewer #1 for providing constructive criticisms throughout this revision and for expressing appreciation of our work.

We have addressed the minor points raised by the Reviewer as follows:

- Line 68, "Shigella" is not normally consider to be a commensal microbe.

The Reviewer is correct in pointing out that Shigella is not considered a commensal bacterium. In fact, our paragraph does not conflict with this notion, but it rather states that a Shigella vaccine strain was engineered to hydrolyze ATP produced by the gut microbiota, which has an inhibitory effect on follicular helper T cells. ATP production, which is the immunoregulatory activity discussed in this section, depends upon the microbiota, and the Shigella vaccine is not a member of it.

- Line 845, it is better to write "16S rRNA gene sequencing".

We agree with the Reviewer and have corrected this line.

- Other things such as figure 1a to figure 1A in line 106 could be fixed during proof reading.

We have corrected this typo.